# Gut Reactions: How Far Are We from Understanding and Manipulating the Microbiota Complexity and the Interaction with Its Host? Lessons from Autism Spectrum Disorder Studies

**DOI:** 10.3390/nu13103492

**Published:** 2021-10-02

**Authors:** Martina Lombardi, Jacopo Troisi

**Affiliations:** 1Department of Chemistry and Biology “A. Zambelli”, University of Salerno, Via Giovanni Paolo II, 132-84084 Fisciano, SA, Italy; lombardi@theoreosrl.com or; 2Theoreo Srl Spin Off Company, University of Salerno, Via Giovanni Paolo II, 132-84084 Fisciano, SA, Italy

**Keywords:** autism spectrum disorder, gut microbiota, gut–brain axis

## Abstract

Autism is a group of neurodevelopmental disorders, characterized by early onset difficulties in social communication and restricted, repetitive behaviors and interests. It is characterized by familial aggregation, suggesting that genetic factors play a role in disease development, in addition to developmentally early environmental factors. Here, we review the role of the gut microbiome in autism, as it has been characterized in case-control studies. We discuss how methodological differences may have led to inconclusive or contradictory results, even though a disproportion between harmful and beneficial bacteria is generally described in autism. Furthermore, we review the studies concerning the effects of gut microbial-based and dietary interventions on autism symptoms. Also, in this case, the results are not comparable due to the lack of standardized methods. Therefore, autism-specific microbiome signatures and, consequently, possible microbiome-oriented interventions are far from being recognized. We argue that a multi-omic longitudinal implementation may be useful to study metabolic changes connected to microbiome changes.

## 1. Introduction

The term “autism” was used for the first time by Bleuler in 1911, in the context of schizophrenia, to indicate a behavior represented by closure, avoidance of the other, and isolation [1].

Subsequently, about 60 years ago, Leo Kanner [2] no longer used this term with the meaning of a “symptom”, but as a descriptive label of the following nosographic entity: infantile autism. Kanner, while stating that it was a congenital condition with unknown etiopathogenesis, in emphasizing the absence of a basic organicity and a peculiar parental type (“refrigerator parents”), opened the way for a psychogenetic interpretation of the disorder. Indeed, in the decades immediately following, the dominant interpretative model was the psychodynamic one, in relation to which autism represented a defense against the anxiety deriving from a failure of the first relations.

Nowadays, different types of autism are considered in the context of a spectrum, using the term “autism spectrum disorder” (ASD). These are united by the deficits in social communication and social interaction, in addition to restricted, repetitive patterns of behavior, interests, or activities [1]. The spectrum nature of ASD means that its symptoms may vary from mild to severe. Some children present disabling language and intellectual problems, requiring lifelong care, while others only show nuanced manifestations. The prevalence of autistic disorder has increased from the 1960s, in which there were 4 cases per 10,000, to 2005, in which there were 10 cases per 10,000 (1:1000). According to current data, the prevalence of ASD would be around 18:10,000. Moreover, when the cases of pervasive developmental disorders, including autism, Asperger syndrome, a pervasive developmental disorder that is not otherwise specified, childhood disintegrative disorder (CDD), overactive disorder associated with mental retardation and stereotyped movements, and Rett syndrome, are considered as a whole, namely, ASD and related disorders, the prevalence estimates would vary around 53:10,000. Males are more frequently affected (M:F = 4:1).

Currently, European and American incidence has shown slightly different values, namely, it affects 1 in 54 children in the USA [3] and 1 in 89 in Europe [4].

Ozonoff et al. [5] reported a ten-fold increase in the risk of developing ASD in children with an ASD-affected sibling. Moreover, Satterstrom et al. [6] showed, in a large exome sequence study, that 102 genetic variations are related to ASD, with a false discovery rate of 0.1 or less. If combined, these results support a combination of non-Mendelian genetic and environmental factors. Furthermore, the immune system seems to be strictly involved [7,8,9,10]. 

An increased expression of genes related to innate immunity [9], as well as low activity of anti-inflammatory mechanisms and a low-grade systemic inflammation [11,12,13], seem to play a role in ASD pathogenesis. 

In this framework, the microbiota role seems to be pivotal, since in early life (in particular, in the firsts 1000 days from birth), immunoregulation development and gut microbiome development are closely linked.

### Gastrointestinal Involvement in ASD 

An intimate relationship between ASD and several medical comorbidities, such as sleep problems and many psychiatry-related comorbidities, i.e., attention-deficit/hyperactivity disorder (AD/HD), anxiety, mood problems, and disruptive behavior, was reported. Anyway, gastrointestinal comorbidities have a special role in their association with ASD. Indeed, since 1943 [2], Kanner reported that ASD subjects showed severe feeding difficulties from their first days of life. Studies related to this association have crossed the entire path of evolution of knowledge on ASD [14,15,16,17]. This association sustained a close relationship between ASD and gut microbiota [18,19,20,21].

## 2. Materials and Methods

A systematic search was performed on Pubmed in April 2021, resulting in 579 peer-reviewed articles that proposed much evidence of gut microbiota and gut–brain axis in autism spectrum disorders. Publications were obtained with the following search query: “autism” OR “autism spectrum disorder” OR “ASD” AND “gut microbiota” OR “gut microbiome”; “autism” OR “autism spectrum disorder” OR “ASD” AND “gut dysbiosis”. Reference lists of reviews and included studies were examined in order to identify potentially relevant articles. Duplicates were manually removed and a total of 449 articles were screened. After careful examination, publications were not included if they (a) were not primarily involved in the corresponding analysis, (b) were not in English, (c) were case reports, (d) were published before January 2016. As a result, 33 articles were included in the qualitative synthesis. The process of study selection is summarized in Figure 1. 

## 3. Gut Microbiome Modification and ASD

### 3.1. Gut Microbiome Modification: Studies and Outcomes 

A growing body of evidence indicates that altered gut microbiota negatively affects neurodevelopment and behavior, suggesting its potential implication in a number of neuropsychiatric disorders [22,23]. 

The idea of a possible involvement of gut microbiota in ASD was firstly postulated in 1998 by Bolte [24], who speculated that *Clostridium tetani* could induce autism. In 2000, a pilot study, published by Sandler et al., reported temporary improvements in both the behavioral and gastrointestinal symptoms in autistic children after six weeks of treatment with oral vancomycin [25]. The gradual regression of the symptomatology, following the discontinuation of the treatment, was explained by a later germination of *Clostridium*’s spores, which are resistant to antibiotics [25,26,27].

These early findings, coupled with the frequent occurrence of GI symptoms in ASD, have led numerous researchers to investigate the composition of the gut microbiota in autistic subjects, comparing it with that of neurotypical controls (NT). 

Starting from those studies, the evidence of microbial dysbiosis in ASD has been growing, although controversial results have been observed among studies. 

In order to describe the current knowledge about the alterations of the gut microbiota in ASD, we compared 21 studies. Their characteristics and key findings are outlined in Appendix A. 

The most abundant phyla reported in the included studies are *Firmicutes* and *Bacteroidetes*, followed by *Proteobacteria* and *Actinobacteria*.

*Firmicutes* were found to be increased [28,29] or decreased [30,31] in autistic subjects, according to different studies. The same occurred for *Bacteroidetes*, with three studies describing higher levels in ASD [31,32,33] and two others reporting the opposite results [28,34]. 

A considerable enrichment has often been described for *Actinobacteria* [28,35,36] and *Proteobacteria* [28,33,36] in ASD. Plaza-Dìaz et al. [36] also performed a subclassification in children with ASD by mental regression (AMR) and no mental regression (ANMR), showing that *Proteobacteria* was only augmented in AMR subjects. 

As for the phyla, heterogeneous results have been observed at the genus level. Among *Firmicutes*, much focus has been given to *Clostridium*, which was found to be increased in autistic subjects compared to controls [31,35,36,37,38,39,40,41,42]. 

These spore-forming bacteria can release pro-inflammatory toxins and potentially toxic metabolites, as indole derivatives [43], p-cresol, and certain phenols [44] that, after reaching the brain through the blood flow, may affect neurodevelopment [45]. Furthermore, some *Clostridium* bacteria have been shown to cause a rise in propionic acid, whose high levels have been correlated to neurotoxic effects [37].

Within *Clostridium* cluster I, the incidence of *C. perfringens,* known to produce a number of toxins and enterotoxins, was significantly increased in the gut of ASD children compared to that in control groups, with the highest levels in ASD subjects with GI symptoms. In particular, recent findings suggest a possible association of β2-toxin gene-producing *C. perfringens* with GI complications in ASD [39,40]. These bacteria have been previously described in gastrointestinal diseases, including sporadic diarrhea, food poisoning, and antibiotic-associated diarrhea [40]. 

A positive correlation between clostridia and GI symptoms in ASD has also been described by Strati et al., who reported a significantly higher abundance of *Clostridium cluster XVIII,* as well as the opportunistic pathogens *Escherichia/Shighella,* in constipated autistic cases [34]. However, further investigation is required to clarify their role in ASD and GI complications. 

Some studies also reported higher levels of *C. difficile* in autistic subjects compared to NT ones [36,41]. Nevertheless, a recent study showed that, although there was a higher percentage of *C. difficile* in autistic cases and their siblings compared to a group of unrelated controls, there was no statistically significant difference between the three groups [27].

Luna et al. [42] reported a considerable increase in several mucosa-associated *Clostridiales* in ASD children with GI disorders, including *Lachnoclostridium bolteae,* which is in compliance with other studies [36,41,42]. On the other hand, other *Clostridiales*, including *Dorea formicigerans* and Blautia luti, as well as Sutterella spp, were decreased [42].

Cao et al. [38] observed an increased abundance of *Desulfovibrio* in ASD, which is in line with previous results. These bacteria could be relevant contributors to GI complications in ASD, as they produce LPS and hydrogen sulfide that can have cytotoxic effects on intestinal cells [22,23,38]. There are still conflicting results about the alterations of the *Bacteroides* genus, which was found to be increased [31,46] or decreased [38,47] in ASD subjects in different studies. Their abundance strongly correlates with the fecal levels of propionic acid (PPA), since they are among the main producers of this metabolite [22,45]. While the high concentration of PPA has been related to behavioral disorders in a number of studies on rodents [45], butyric acid, another relevant short-chain fatty acid (SCFA), is known for its anti-inflammatory and protective properties. Indeed, it can protect the integrity of the intestinal epithelial barrier, strengthen mucosal immunity [30,37], and also modulate neurotransmitter gene expression [32,37]. Furthermore, it has been reported that butyrate can restore blood–brain barrier permeability by inducing an increased expression of tight junction proteins, supporting its essential role in the physiological activities of the gut–brain axis [30]. 

Several studies reported significant alterations in the relevant abundance of butyrate producers in ASD subjects, such as various members of *Lachnospiraceae* and *Ruminococcaceae* families, whose levels have been found to be lower in ASD cases compared to controls [30,36,37,38]. 

Decreased levels of *Bifidobacterium*, which can have beneficial effects through its anti-inflammatory properties, have often been reported in ASD [33,38,46]. Furthermore, Ahmed et al. discovered that the only significant difference between the gut microbiome of autistic children and that of their healthy siblings was the higher abundance of *Bifidobacterium* in the siblings’ group, supporting its protective role [46]. However, it must be said that other studies reported the opposite trend between autistic and neurotypical subjects [28,29,36]. Interestingly, higher levels of *Lactobacillus*, widely recognized as probiotics, have been reported in ASD [29,34], but, as for other bacteria, there is no consensus among the studies [48]. 

Several studies also described a decreased abundance of *Prevotella*, a commensal microorganism that plays an important role in saccharides metabolism and the biosynthesis of vitamins [28,29,34,49]; lower levels have also often been reported for *Veillonella* [32,34] and *Faecalibacterium* [29,47,49]. Ding et al. also observed an association between the severity of symptoms and the relative abundance of *Faecalibacterium* strains, with the lowest levels in children with severe ASD compared to those with mild ASD and healthy controls. The opposite trend was described for unidentified *Lachnospiraceae* and *Erysipelotrichaceae* strains [47].

Another considerably decreased bacterial genus is *Akkermansia*, especially the *Akkermansia muciniphila* species, and, as they are crucial mucin degraders, their reduction may result in a relevant increase in gut permeability [31].

Even though most of these studies were focused on bacteria, a few of them also reported alterations in fungal components of the gut microbiota. At the phylum level, no statistically significant differences were detected. At the genus level, two studies reported a relevant increase in the abundance of *Candida*, especially in the *Candida albicans* species [34,48]. The release of ammonia and toxins, as well as the reduced absorption of minerals and carbohydrates, due to the increased counts of *Candida*, may lead to autistic behavior [48,50]. These fungi commonly colonize mucosal surfaces of the GI tract, where their growth is strictly regulated through competition with and suppression of the resident flora [48]. Indeed, gut bacteria and fungi live in a subtle balance and mutually influence each other. It has been observed that bacterial dysbiosis after antibiotic treatments can lead fungal commensals to bloom. As a consequence, the colonization of *Candida albicans* can interfere with the restoration of the healthy bacterial community, further contributing to dysbiosis [23,34]. 

In contrast with previous results, a recent study reported higher levels of *Candida albicans* in controls compared to ASD subjects. They also detected an increase in *Saccharomyces*, potential human pathogens, and a lower abundance of *Aspergillus* in ASD. In particular, the highest abundance in autistic subjects was found in *Saccharomyces cerevisiae*, whose high levels have already been observed in schizophrenic patients. Among *Aspergillus,* the *Aspergillus versicolor* species, well known for its metabolites with anti-inflammatory activities, was significantly decreased in autistic subjects [50].

Overall, several studies reported an increase in the abundance of harmful bacteria and a decreased presence of beneficial ones, but observations on individual microbial taxa are often contradictory and, currently, it is not possible to describe a specific microbial signature of ASD. 

These discrepancies may be attributed to several reasons, including the restricted number of participants and the considerable differences in sampling methods, analytical techniques, referred databases, and statistical methods among the studies. The aforementioned factors negatively impact the reliability of the comparison between studies, suggesting the need for standardized methods of analysis. Furthermore, the composition of the gut microbiota can be significantly affected by geographical, dietary, genetic, environmental, and cultural differences, which should be considered as additional confounding elements. For these reasons, all the results must be considered with caution. 

### 3.2. Gut Microbial-Based Treatments 

In the past two decades, the mechanisms underlying the bidirectional communication between the GI tract and the brain, through the so-called gut–brain axis, have been a subject of fast-growing interest. While the brain–gut interactions have been extensively investigated, the role of the gut microbiome as a key modulator of brain health and disease has only recently been addressed [51]. Although further studies are needed in order to fully elucidate its implications and signaling pathways within this context, it is clear that the gut microbiome can affect brain activities, both directly and indirectly, through a number of neural, endocrine and immune connections [52]. Ever since the influence of the gut microbiome on brain functions has been proved, many efforts exploring the impact of dysbiosis on neuropsychiatric disorders have been performed. Gnotobiology studies on germ-free (GF) animals demonstrated that the brain and behavior are significantly affected in the absence of a gut microbiome [53,54,55]. Furthermore, the transplantation of an ASD microbiome into GF mice induced altered behaviors, including decreased sociability and repetitive behavioral patterns [56,57], as well as alternative splicing of ASD-relevant genes, suggesting that a pathogenic microbiota may contribute to the genesis and development of the disease [56]. In light of the above, it is reasonable to speculate that targeting the gut microbiome may be a novel and safe therapeutic approach for ASD. 

In order to describe the recent evidence regarding the effects of gut microbial-based interventions on ASD, we analyzed 12 studies. The characteristics of the included studies are provided in Appendix A. 

The most investigated approach is the administration of probiotics, sometimes combined with prebiotics. The International Scientific Association for Probiotics and Prebiotics (ISAPP) defines probiotics as “*live microorganisms that, when administered in adequate amounts, confer a health benefit to the host*” and prebiotics as substrates that are “*selectively utilized by host microorganisms, conferring a health benefit*” [58]. The use of probiotics in the management of a psychiatric disorder was first proposed by Logan and Katzman, who postulated that the administration of probiotics may be used as an adjuvant therapy in patients suffering from major depressive disorder [59]. In 2013, Dinan and colleagues used the term “psychobiotics” to describe a novel class of probiotics with potential therapeutic applications in treating psychiatric illnesses [60]. Even though the exact mechanisms by which these microorganisms act have not been clarified yet, a growing body of literature provides evidence of their beneficial effects on ASD. However, the different studies evaluated the effects of distinct probiotic mixtures. 

A supplement formula based on three probiotic strains, including *Lactobacillus acidophilus*, *Lactobacillus rhamnosus,* and *Bifidobacterium longum*, was administered to a group of ASD-diagnosed children for three months. Significant improvements in both the GI and core symptoms of ASD were observed after the treatment, as well as a notable amelioration of gut microbiome composition, revealed by a substantial increase in Bifidobacteria and Lactobacilli [61]. De Simone Formulation (DSF), marketed as Visbiome^®^ in the USA and Vivomixx^®^ in the EU, is a probiotic supplement made up of eight probiotic strains, mostly Lactobacilli and Bifidobacteria, including *Lactobacillus para-casei*, *Lactobacillus plantarum*, *Lactobacillus acidophilus*, *Lactobacillus delbrueckii* subsp. *Bulgaricus, Bifidobacterium longum*, *Bifidobacterium infantis*, *Bifidobacterium breve,* and *Streptococcus thermophilus.* In a double-blind randomized controlled trial by Santocchi et al., DSF was orally administered to a group of ASD children at the posology of two packets/day for one month and one packet/day for the following five months. According to their findings, autism severity, as well as the levels of plasma and fecal inflammatory biomarkers, did not show any statistically significant difference after the treatment. Interestingly, the supplementation with DSF, compared with a placebo, provided different effects on ASD subjects with and without GI symptoms. Indeed, the children of the first subgroup experienced improvements in their GI complaints, sensory profiles, and adaptative functioning after the treatment [62]. These results are in line with a previous pilot study by Arnold et al., who described an amelioration in GI symptoms in a cohort of ASD children treated with DSF [63]. On the other hand, children without GI symptoms showed significant modifications in their core symptoms of ASD, revealed by a decline in ADOS-CSS scores (total autism diagnostic observation schedule—calibrated severity score), compared with a placebo. These observations suggest that probiotics could positively affect the behavioral symptoms of ASD independently from the intermediation of their effects on GI symptoms. Furthermore, children with and without GI symptoms may potentially represent distinct populations in which probiotics could act through different mechanisms, probably due to different microbiota targets [62]. 

In 2020, Wang et al. compared the gut microbiota composition, the fecal SCFAs, and the plasma neurotransmitters of 26 autistic subjects with those of 24 neurotypical controls. Then, they administered a group of 16 ASD patients with a probiotics+FOS supplementation and 10 others with a placebo supplementation for 30–108 days. The mixture supplied to the probiotics+FOS group was composed of fructo-oligosaccharides and four different probiotic strains, including *Bifidobacterium infantis* Bi-26, *Lactobacillus rhamnosus* HN001, *Bifidobacterium lactis* BL-04, and *Lactobacillus paracasei* LPC-37. Consistently with previous studies, the gut microbiota profiling revealed a significant dysbiosis in children with ASD, with higher levels of potentially pathogenic bacteria, such as *Clostridium* and *Ruminococcus*, and lower counts of beneficial ones, including *Bifidobacteria* and especially *B. longum*. The fecal levels of SCFAs, particularly propionic acid, butyric acid, and acetic acid, were found to be considerably lower in ASD subjects compared to NT controls, suggesting a notable reduction in the fermentation capacity of their microbiome. Furthermore, plasma neurotransmitters and the related metabolites were significantly imbalanced in ASD subjects, who showed a conspicuous hyperserotonergic condition and metabolic disorders in dopamine metabolism and the tryptophan–kynurenine pathway. The effects of the supplementation were longitudinally assessed at 0, 30, 60 and 108 days from the start of the treatment. While the ASD subjects of the placebo group did not exhibit any change over time, the probiotics+FOS counterpart showed significant improvements in both GI and core symptoms of ASD, and the beneficial effects were more prominent with the time of administration. In terms of gut microbiome composition, the probiotics+FOS intervention resulted in a significant decrease in *Clostridium*’s relative abundance and an increase in *B. longum.* SCFAs levels gradually improved, approaching values similar to those in the control group, and the same happened for plasma neurotransmitters and the related metabolites. As part of this study, zonulin was used as a marker of intestinal permeability, and its plasma levels were found to be significantly higher in ASD subjects compared to NT controls at baseline. After the probiotics+FOS intervention, these levels decreased, suggesting that the employed supplementation could also lead to an amelioration of the leaky gut [64]. In addition, previous studies described a possible link between *Lactobacillus* levels, ASD-related behaviors, and the GABAergic system, and suggested that these bacteria may regulate the expression of GABA receptors in the brain by secreting GABA, the most important CNS inhibitory neurotransmitter [65]. Consistently with those observations, Tabouy et al. found that the administration of *L. reuteri* to Shank3 KO mice resulted in increased GABA receptor expression in the brain, coupled with a partial attenuation of unsocial behaviors in male mice, and repetitive patterns in both male and female mice [66]. Overall, the probiotic interventions were well tolerated and no serious adverse events were observed. Despite the considerable heterogeneity of the studies, in terms of composition, duration and concentration of the treatments, the above findings suggest that probiotic supplementation may be an effective and safe treatment for ameliorating ASD-related symptoms. However, while probiotic interventions supplement only a few bacterial strains, fecal microbiota transplantation (FMT) can ensure the transfer of approximately a thousand bacterial species. This treatment, whose origins date back to the fourth century China [67], is widely used in the case of recurrent and refractory *Clostridium difficile* infections, and consists of the transfer of gut bacteria, obtained from a stool sample of a healthy donor, to a recipient, in an attempt to improve and possibly normalize the whole microbial microenvironment. As a consequence, FMT carries the risk of transmitting infections and transferring pathogens, so a thorough screening of donors is required prior to donation, in order to reduce and prevent the occurrence of adverse events [68,69]. In 2017, Kang et al. evaluated the effects of a modified FMT protocol, called microbiota transfer therapy (MTT), on the gut microbiome composition, and GI and core symptoms of ASD in an open-label trial involving 18 autistic children with moderate-to-severe GI dysfunctions. The participants received a 2-week vancomycin treatment, a bowel cleanse, and then they were orally or rectally administered with a high dose of standardized human gut microbiota (SHGM), followed by daily lower maintenance oral doses for 7–8 weeks. In order to define if the effects of the treatment were temporary or long lasting, clinical responses and gut microbiota composition were monitored for an additional 8-week follow-up period. Both ASD-related behaviors and GI symptoms, especially abdominal pain, indigestion, diarrhea, and constipation, were found to be significantly improved at the end of the treatment. Furthermore, MTT resulted in a substantial increase in gut bacterial diversity and in the counts of beneficial bacteria, including *Bifidobacterium*. All the beneficial effects were maintained after 8 weeks from the end of the treatment and no serious adverse events were described [68]. Two years later, the same 18 subjects were re-evaluated in a follow-up study, in order to investigate the long-term impact of MTT on their symptoms and gut microbiome. By performing the same GI and behavior tests that were employed in the original trial, it was observed that the reduction in GI symptoms persisted, and the severity of ASD was slowly and steadily improved by the end of the treatment. In addition, they described a positive correlation between improvements in GI and the core symptoms of ASD, supporting the hypothesis of a clinical link between GI health and behavior. Furthermore, the alterations in gut microbiome composition, with higher levels of *Bifidobacterium* and increased bacterial diversity, persisted after two years [70]. Beneficial effects of FMT on ASD have also been described in murine models. Indeed, Goo et al. observed that FMT from normal to *Fmr1* KO mice significantly ameliorated autistic-like behaviors, especially cognitive dysfunctions and social withdrawal behaviors [71]. These results suggest that FMT may be a promising approach to treat ASD-related symptoms, but further investigations are needed in order to establish the optimal protocol and donor type. However, it is important to note that the clinical applications of FMT can be challenging, due to the laborious and expensive donor selection, and the heterogeneity of the donor’s microbiome, which can lead to different effects on the recipients. On the other hand, gut microbiome transplantation (GMT) with an in vitro cultured microbiome may be an advantageous alternative to circumvent these limits, since the procedure is faster, easier, and repeatable. This type of transplantation resulted in significant improvements in anxiety-like behavior, stereotyped behavior, and gut microbiome composition in maternal immune activation (MIA) mice offspring [72], but, to the best of our knowledge, this is the only study evaluating its effects on ASD; thus, further rigorous trials are needed.

Moreover, some authors evaluated the effects of dietary interventions on the gut microbiome, GI and core symptoms of ASD. Indeed, autistic patients often exhibit difficult eating behaviors, including food selectivity and repetitive eating patterns, which, besides implying risks of malnutrition, can deeply affect the gut microbiota composition [73]. The ketogenic diet (KD) is a high-fat, low-carbohydrate, adequate-protein diet, which is already an established treatment for pharmacologically resistant childhood epilepsy [74] and has been suggested to have neuroprotective effects in ASD [75]. Since previous studies found that KD was able to improve ASD symptoms in BTBR mice [76], Newell et al. investigated whether this diet was able to modify their gut microbial profiles. Interestingly, BTBR mice fed a KD for 10–14 days exhibited a considerable decline in the total bacterial abundance in both their cecal and fecal matter, probably due to the reduced presence of undigested carbohydrates, which are normally metabolized by the gut microbiome. Furthermore, KD mitigated the increase in Bacteroidetes and the decrease in Firmicutes assessed at baseline, and led to a notable augmentation of the saccharolytic SCFA-producers *C. coccoides* and *C. leptum*, suggesting that the benefits of this diet on ASD symptoms may potentially be attributable to its effects on the gut microbiome [77]. In addition to KD, the impact of specific nutritional supplements, such as prebiotics and vitamins, has been investigated in ASD subjects. A 6-week Bimuno^®^ galacto-oligosaccharide (B-GOS^®^) prebiotic supplementation was found to have beneficial effects on behavior, gut microbiome composition, and metabolic profiles in a group of autistic children on unrestricted diets. More specifically, the treatment led to a notable increase in Bifidobacteria, especially *B. longum*, and in members of the Lachnospiraceae family, widely recognized as butyrate producers. Furthermore, the fecal levels of amino acids (AA) were decreased after the treatment [78] and, since high stool levels of AA have been previously associated with an altered gut barrier [79], this reduction suggests that B-GOS^®^ supplementation may also improve gut health. Notable alterations in gut microbiome composition were also described in a cohort of autistic children administered with vitamin A supplementation, but no statistically significant differences in ASD symptoms were observed after the treatment [80]. Previous studies described an association between ASD and suboptimal breast-feeding practices, including non-intake of colostrum and bottle-feeding [81]. In a recent pilot study, Sanctuary et al. found that bovine colostrum product (BCP), both administered as its own treatment or in combination with *B. fragilis*, was able to ameliorate GI symptoms and aberrant behaviors, including stereotyped behavior and hyperactivity, in a cohort of autistic children [82]. BCP is an important source of immune factors, including immunoglobulin, lactoferrin, and cytokines, which can impact gut microbial composition and the immune system [83]. In addition, it contains complex milk oligosaccharides, which are the third most abundant components of human milk and are known to promote the growth of beneficial bacteria, especially Bifidobacteria [84]. However, no significant differences in gut microbial composition were observed in this study [82]. 

Together, these data provide suggestive, but not conclusive evidence about the effectiveness of gut microbial-based interventions in alleviating ASD symptoms (Figure 2). Indeed, most of the included studies were affected by several methodological weaknesses, including a restricted sample size and a lack of reliable laboratory indicators to quantitatively evaluate the efficacy of the treatments. Therefore, since the changes in gut microbial composition result in metabolic differences, the use of multi-omics is strongly recommended for future research. Overall, further investigations are needed in order to determine the long-term effects of these treatments, as well as to establish which method and which bacteria can provide benefits in autistic patients. 

## 4. Discussion

### Perspectives from Ongoing Investigations

Large differences in both microbial composition and gut microbiome modification treatment outcomes were reported in ASD subjects. Several hypotheses were reported to justify them. First of all, the intrasubject variability was very large, decreasing the chance to individuate a consistent trend and/or specific dysregulation that could be certainly related to the disease signature. Moreover, further increasing this variability, the wide age-related microbiome modifications were often poorly considered. Furthermore, no standards were currently recognized and applied in microbiome-related studies, exceptionally increasing the variability of the reported results. Microbiome analysis standardization is a pivotal step to try to manage these limitations. The European Commission has pushed for these standardizations since 2011 by promoting, by means of the Seventh Framework Programme, the building of the International Human Microbiome Standards (IHMS) (http://www.human-microbiome.org/, accessed on 10 August 2021). The Consortium released several standard operative procedures [85,86,87,88,89] that could be applied for sample collection, nucleic acids extraction and sequencing, and data analysis. The Consortium has been recently awarded a Horizon 2020 grant [90] within the project of the International Human Microbiome Concertation and Support Action (IHMCSA), whose aim include “*to tackle all necessary steps to open the perspective of managing nutrition and health of the microbial human building consensus on priorities and means for the future of microbiome science and its translation*”. In particular, it will promote unified repositories for sharing standards, SOPs, and data, and will contribute to the structuration of the European Microbiome Centers Consortium with a role in gathering world microbiome networks of excellence. The hope is that this would increase the reproducibility and comparability of the data, also increasing their meaning values.

Another key issue that should be solved concerns the disambiguation between the causality and pure effect of the reported microbiome signatures. Case-control studies do not allow an understanding of if the reported dysbiosis are linked to disease onset, or if it is just a consequence of it. In particular, although several mechanistic interpretations were proposed for many over or under gut microbe representations, the most part of the reported experimental designs do not allow an understanding of if these could be the results of the different social behaviors or food habits of the ASD subjects. 

Again, the European Commission, showing great attention to these aspects, promotes the creation of an international consortium, whose aim is to longitudinally investigate the differences in microbiome evolution in children who develop ASD. To do so, the GEMMA consortium [91] will follow 600 at-risk babies who were born from parents who already raise an ASD-affected son. The hope is that the analysis of the evolution of the gut microbiome, corrected to all the possible confounders (environmental effects, drug taken, genetic effects, and so on and so forth), will allow a specific factor(s) to be described (or some) that will start a cascade of events, which eventually culminate in the disease onset. Therefore, the next few years will be crucial, and will probably open new horizons on understanding the delicate and complex relationship between humans and microbes. 

## 5. Conclusions 

In the past few years, considerable advances in understanding the importance of the gut microbiome in brain health and disease have been performed, with more and more authors suggesting its implications in a number of neuropsychiatric disorders. Several studies reported significant alterations in gut microbiome composition in patients suffering from ASD, who exhibit higher levels of potentially harmful bacteria and lower counts of beneficial ones. Nonetheless, currently, it is not possible to describe a unique microbial signature of these disorders, due to the lack of standardized methods of analysis, as well as the notable heterogeneity of participants’ characteristics. However, the accumulating evidence from human and animal research suggests that gut microbial-based treatments may be useful as a new and safe therapeutic approach for ASD. Further investigations are needed in order to fully elucidate the role of the gut microbiome in the pathogenesis of autism and its potential therapeutic applications. The use of multi-omics is strongly recommended for future research, in order to obtain more conclusive results. 

## Figures and Tables

**Figure 1 nutrients-13-03492-f001:**
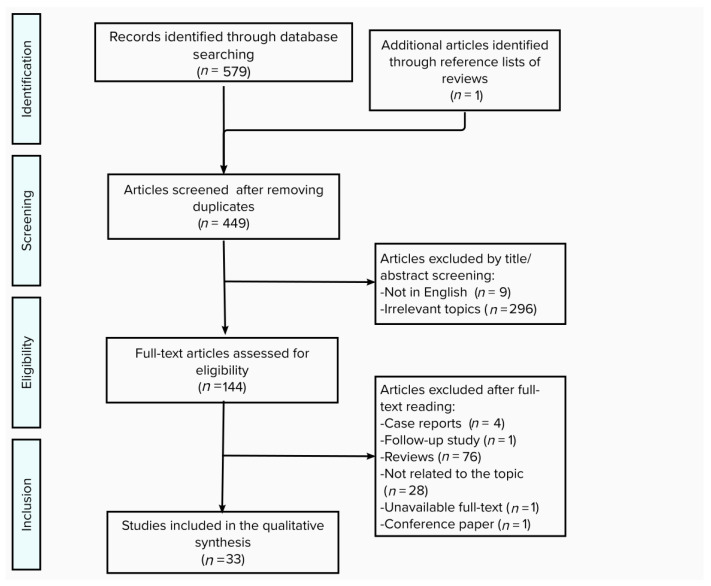
Flow diagram summarizing the selection of studies.

**Figure 2 nutrients-13-03492-f002:**
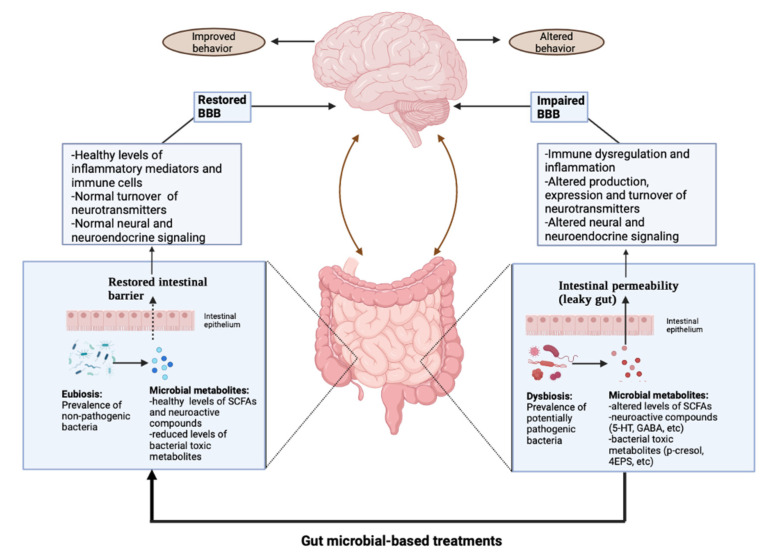
Potential effects of gut microbial-based interventions.

## Data Availability

Not applicable.

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
