# Peer review of "Gut Reactions: How Far Are We from Understanding and Manipulating the Microbiota Complexity and the Interaction with Its Host? Lessons from Autism Spectrum Disorder Studies"

_nutrients, 2021, doi:10.3390/nu13103492_

Round 1

Reviewer 1 Report

The authors have done an excellent job on both the articles reviewed, but also the gaps that remain in the literature. This paper serves to create a translational bridge between Pediatrics, Neurology, Gastroenterology, and Microbiology. This relationship will only be made stronger by thoughtful systematic analysis. 

I found two typographical errors for editors to correct:

Line 439: multiple space gap between of and it

Line 449: genetic and effects should be separated

Author Response

We are thankful to the reviewer for the comment and the useful improve suggests. We provided a new MS version accounting for all the suggested improvements.

Reviewer 2 Report

Dear Authors,

The Editors of Nutrients have kindly invited me to review of your manuscript. Please find my comments below: 

  • The title should go "How far are we", as this is a question. 
  • Line 37: "Nowadays, different types of autism are considered" (not "autisms").
  • Line 67: Please correct typo. "intimate relationship". 
  • Section 2. Materials and methods. Great section. 
  • Section 3. Gut microbiome modification and ASD. Beautifully written providing excellent insights from the literature review. 
  • Figure 2: I would recommend using "Intestinal permeability (leaky gut)" in the right section of the diagram. 
  • Section 4 should be a "Discussion", with 4.1 being "Perspectives from ongoing investigations", and 4.2 a brief conclusion (no more than 15-20 lines).

I hope you find my comments useful in the review process of your manuscript, which I have found most enlightening. If you were able to weave in my recommendations, I should be only too happy to recommend that your review be accepted for publication. 

Kind regards,

The reviewer 

Author Response

(The authors gave the same response as above.)
